# The Critical Role of miRNAs in Regulation of Flowering Time and Flower Development

**DOI:** 10.3390/genes11030319

**Published:** 2020-03-17

**Authors:** Saquib Waheed, Lihui Zeng

**Affiliations:** 1College of Horticulture, Fujian Agriculture and Forestry University, Fuzhou 35002, China; swaheed022@gmail.com; 2Institute of Genetics and Breeding in Horticultural Plants, Fujian Agriculture and Forestry University, Fuzhou 350002, China

**Keywords:** microRNAs, flower regulation, juvenility, floral induction, flower development

## Abstract

Flowering is an important biological process for plants that ensures reproductive success. The onset of flowering needs to be coordinated with an appropriate time of year, which requires tight control of gene expression acting in concert to form a regulatory network. MicroRNAs (miRNAs) are non-coding RNAs known as master modulators of gene expression at the post-transcriptional level. Many different miRNA families are involved in flowering-related processes such as the induction of floral competence, floral patterning, and the development of floral organs. This review highlights the diverse roles of miRNAs in controlling the flowering process and flower development, in combination with potential biotechnological applications for miRNAs implicated in flower regulation.

## 1. Introduction

The vegetative-to-reproductive phase transition is a very critical and distinctive change which undergoes in plants during their life cycle. In order to flower, plants have to go through different morphological and physiological changes under dynamic environmental conditions. Under normal circumstances, external cues such as low temperature (vernalization) and light (duration of exposure and intensity) are the prime factors that determine when plants will blossom [1]. The endogenous cues such as plant age, phytohormones (mainly gibberellic acid) and carbohydrate status (mainly sucrose) coordinate with external signals to determine the appropriate flowering time. In plants, multiple regulatory pathways involving the internal and external signals governing the flowering process have been postulated. Amongst the environmental factors photoperiod or light is a decisive factor in determining the flowering time. Based on photoperiod, plants have been classified in three categories which are short-day plants (long-nights), e.g., rice; long-day plants (short-nights), e.g., *Arabidopsis*; and day-neutrals, e.g., tomato [2]. In recent studies, many circadian clock-related genes have been identified which specifically play a role in regulating the flowering time in response to different photoperiods. Temperature is another critical factor involved in flowering time determination. Therefore, plants behave differently due to seasonal changes in temperature. In many plants flowering needs a prolonged period of cold temperature which is termed as vernalization followed by exposure to normal ambient temperature [3]. Other major endogenous factors that regulate the flowering process are the appropriate age of the plant, sugar assimilates and gibberellin signaling [1]. Collectively, endogenous and exogenous factors contribute to regulating five major floral transition pathways, namely; the photoperiod pathway, the vernalization pathway, the gibberellin pathway, the age pathway, and the autonomous pathway. However, the endogenous regulators are independent of vernalization and photoperiod. These pathways form an integrated regulatory network to crosstalk with each other and channelize the signals via several floral integrators to regulate floral transition in plants [4]. In some plants other than these major pathways, plant hormones such as abscisic acid [5], cytokinins [6], ethylene [7], and brassinosteroids also contribute to the flowering process [8]. Gibberellin interacts with other hormonal pathways as well.

MicroRNAs (miRNAs) play essential roles in flowering due to their role in post-transcriptional genetic regulation. These are 21–22 nucleotide-long noncoding RNAs and have been studied extensively and characterized in different plant species. These are highly conserved across species within each of the plant and animal kingdoms but no individual miRNA shares sequence similarity between the two lineages [9]. MiRNAs were discovered in the early 1990s as the product of a gene that controls developmental timing in *Caenorhabditis elegans* [10]. However, it was not until 2001 that miRNAs were shown to be broadly distributed in the animal kingdom [10,11]. Shortly after that, numerous miRNA families were discovered in the plant kingdom [12,13]. Many plant miRNAs are evolutionarily conserved, and play a vital role in the regulation of numerous essential developmental processes such as shoot apical meristem (SAM) regulation, leaf and root system regulation, organ development and plant floral transition [14,15,16]. In recent years, various reports signify the importance of miRNAs in floral transition and flowering time control by regulating the expression of the genes involved in these processes [17]. Research on the model plant *Arabidopsis* has provided an adequate amount of evidence, and results obtained from other plants such as rice, maize, potato and other species demonstrated the conservation of miRNAs and their integrated regulatory pathway/gene network [17,18,19].

In plant biology, the scientific literature regarding the role of miRNAs in plant development has increased significantly in recent years [20,21,22]. Many researchers have contributed some interesting and valuable review articles in this aspect focusing on the role of miRNAs in flower regulation and the development of single or multiple plant organs [23,24]. Therefore, we think that there is a need for a comprehensive and critical review elaborating on the mechanisms and actions of plant miRNAs involved in the control of flowering time and floral organ development. This review is a detailed discussion on current knowledge and recent progress regarding the miRNA families and their involvement in floral regulation and development.

## 2. Overview of miRNA Processing and Functioning

The discovery of miRNA is an added tool in the plant genetics arsenal for the genetic improvement of crops. The genetic expression can be regulated by using miRNAs that can target specific mRNAs. Before describing the function of various miRNAs involved in flower regulation and development, we will first briefly review how plant miRNAs are produced and molecular details of their function.

### 2.1. Biogenesis of Plant miRNAs 

miRNA biogenesis is well understood in animal miRNAs. However, this process is still not very familiar when it comes to plants [25]. miRNAs are encoded in many loci in the intergenic regions of the plant genome [26]. Most eukaryotic miRNA genes are RNA polymerase II (Pol II) transcription units, but studies showed that some miRNAs are also transcribed by RNA polymerase III [27]. In the process of biogenesis, RNA Pol II transcribes primary miRNA (primiRNA), which contains many features of RNA poly II transcripts such as 5’ cap and 3’ poly (A) tail. Pri-miRNA with stem-loop structure is stabilized by RNA-binding protein Dawdle (DDL) before it is processed into premature miRNA (pre-miRNA) in the nuclear processing center (D-body) (Figure 1). Conversion of pri-miRNA to pre-miRNA is orchestrated by functions and interactions of DCL proteins, Hyponasty Leaves (HYL1), Serrate (SE) and nuclear cap-binding complex (CBC). It is believed that the DCL protein also functions in the processing of miRNA-miRNA^*^ (~21 nt) from pre-miRNA since plants lack a Drosha-like enzyme [28]. 

The miRNA-miRNA^*^ duplex with two nucleotides overhanging at 3’ ends is methylated by Hua Enhancer 1 (HEN1) to protect it from being degraded by Small RNA Degrading Nuclease (SDN) exonuclease. The miRNA-miRNA^*^ duplex is then exported to the cytoplasm by one of the Exportin 5 ortholog HASTY and later integrated into RNA-induced silencing complex (RISC), a multiprotein complex of ribonucleoprotein (Figure 1) [27]. Nonetheless, only a single strand becomes a mature miRNA while the other strands are eventually degraded by an unknown mechanism [27]. 

### 2.2. miRNA-Mediated Regulation of Gene Expression

Small RNAs are synthesized by RNA dependent RNA polymerase from exogenous RNA genome or derived from the transcription of inverted-repeat encoded in endogenous loci [29]. Upon dicing and modifying by effector proteins, the small RNAs can be engaged in transcriptional gene regulation by modifying chromatin activity in the nucleus, or they are exported to the cytoplasm to induce post-transcriptional gene expression (PTGS) [30]. In terms of target recognition and function, plant miRNAs must be complementary to their targets to regulate gene expression [31]. Generally, plant miRNAs recognize their targets in the region of miRNA responsive elements (MRE) which are located in the coding regions [32], 5′ untranslated region (UTR) or 3′ UTR [33] of the targets. 

miRNA-induced PTGS is accomplished by the functions and interactions of Argonaute (AGO) proteins in RISC and other unknown factors [27]. The miRNA itself does not has the ability to cleave mRNAs or interfere with translation of the targets but it plays roles in scanning the appropriate target. To be functional, mature miRNA is incorporated into RISC containing one member of the AGO protein family. The AGO protein contains a small RNA-binding PAZ domain and PIWI domain with catalytic residues that confer endonucleolytic activity to that RISC [27]. Upon binding to the appropriate target, RISC induces the cleavage of the target transcripts [27]. As cleavage usually occurs in the middle of the target transcripts, it was suggested that RISC cleaves the target transcripts in the region where miRNA and targets are complementary. However, recent studies have shown that cleavage sites can also be found beyond these miRNA complementary regions [34].

In addition to cleaving the target, plant miRNAs can also regulate gene expression through translational inhibition of the targets [29]. In recent studies, it has been observed that sequence binding of miR156 and miR157 with their targets, SQUAMOSAPROMOTER BINDING PROTEIN-LIKE 3 (*SPL3*) gene at 3′-untranslated region (UTR) renders translational repression rather than degradation of the transcript [35]. Similarly, another study showed that the expression of a floral homeotic gene APETALA2 (*AP2*) is regulated by miR172 through translational inhibition of the transcript [36]. Thus, it is evident from the fact that plant miRNAs regulate gene expression through cleavage or translational repression of the targets. It is also believed that plant miRNAs employ both mechanisms to regulate gene expression. Nevertheless, the prevalence of one mechanism based on the position or degree of base pairing between plant miRNAs and their targets is unpredictable based on current knowledge in the field.

### 2.3. miRNAs Regulate Transcription Factors

The majority of plant miRNAs regulate the genes encoding for transcription factors (TFs), though some are also involved in the regulation of other genes function in plant immune response, gibberellins signaling and floral transition. It is unclear why miRNAs preferentially target TFs but these two types of regulators share common actions in the regulation of their target genes. It has been reported that TFs and miRNAs are the largest families of gene regulatory molecules in multicellular organisms [37], miRNAs regulate gene expression at the post-transcriptional level, but TFs act at the transcriptional level. However, these regulators are similar in terms of pleiotropic effects, and degree of accessibility to their binding sites in the targets. An individual TF and miRNA can regulate dozens of target genes; conversely, many individual genes are found to be regulated by many TFs and miRNAs [37,38]. Although TFs are functional proteins, they still require cofactors and enhancers to bind the target DNA sequences and to operate the functions. Hence, both TFs and miRNAs require combinatorial and cooperative activity from effector complexes. Besides, both regulators have been restricted to their target binding sites. Studies have shown that most plant miRNA targets are TFs that play crucial roles in plant development and floral transition. Sun et al. (2012) summarized a list of miRNAs and their targets and validated their possible functions in plants [29]. Which includes embryonic development, leaf morphogenesis, shoot branching, root branching, flowering time, signal transduction and response to biotic and abiotic stresses [29]. In rice computational analysis of fully sequenced genome revealed that 46 genes were predicted to be the targets of new miRNAs, while 16 genes encoded the TF [39]. Similarly, it was found that 12 out of 26 miRNA families in maize regulate TF genes [26]. 

### 2.4. MicroRNAs and Plant Development

The recent identification of an increased number of plant miRNAs and their target genes encoded TFs as well as their widely accepted role in plant developmental patterning, demonstrates the significance of miRNAs in plant developmental regulation [40]. Over the past decade, miRNAs targeting TF genes have shown to regulate many aspects of plant biology, such as hormone response, metabolism, biotic, and abiotic stress [27]. The role of miRNAs in plant development is evident from the phenotype of mutants defective in miRNA biogenesis, miR168 regulates the expression of AGO genes, especially miR168b-regulated AGO1 plays a crucial role in plant development [41], since miR168 is involved in regulating the key component of RISC, any variation in miR168 level has a potential effect on the regulatory action of other miRNAs. In *Arabidopsis*, miR164 targets the TFs known as cup-shaped cotyledon (*CUC*) and No Apical Meristem (*NAM*), which are involved in root and shoot development [42]. The miR164 negatively regulates the genes that encode the *NAC* domain ‘TF’ such as *CUC1* and *CUC2*, which are necessary for the formation of boundaries between meristem and emerging organ primordia in *Arabidopsis* [43,44]. It was reported that miR164 mediated regulation is also necessary for the proper formation of organs [45]. The miR165 is involved in HDZIP-III mediated indeterminacy in apical and vascular meristems [46]. 

The plant leaf is the primary photosynthesizing organ and plays an essential role in plant growth and productivity. A study on tomato showed that the ectopic increase of miR319 could significantly alter the leaf size and shape [47]. Similarly, overexpression of miR319 in rice and creeping bentgrass results in a wider leaf blade [48,49]. Furthermore, miR319 not only participates in leaf development but also in shoot and floral organ growth. A recent study showed that miR396 could regulate GROWTH REGULATING FACTORS (*GRFs*) genes and their up-regulation in *Arabidopsis* leads to dramatically enlarged cotyledons and leaves [50]. The overexpression of miR396 in *Arabidopsis* remarkably represses expression of *GRFs*, thereby causing narrow-leaf phenotypes [51].

miR156 and miR172 are the most ancient miRNAs [15,52], which play crucial regulatory actions in vegetative to reproductive phase transition of plants. Overexpression of miR156 presents a prolonged vegetative phase and late-flowering phenotypes [53,54,55,56]. Whereas plants overexpressing miR172 resulted in accelerated flowering phenotypes [57]. It is well-known that altered regulation of miR156-targeted SPLs display altered phenotypes in tomato, including the different number of leaves, semi-dwarfed size and retain longer vegetative phase [58]. The altered expression of miR393 in *Arabidopsis*, *Medicago truncatula* and *Oryza sativa* showed improved stress tolerance against low temperature, salinity and drought conditions [59,60,61].

Recent discoveries suggest that miRNAs have appeared as a crucial regulator of hormonal signaling pathways through affecting their metabolism and distribution in plants. Hormone signaling pathways play an essential role in coordinating multiple developmental programs including the germination process, organ morphogenesis, and floral transition in response to various environmental conditions [62]. The latest studies demonstrated that some miRNAs act as a key regulator of auxin signaling pathway and directly or indirectly affect floral transition in different plant species [63,64,65]. Similarly, miR390 and miR393 are involved in controlling the auxin signaling pathway to influence such actions of plants [66,67,68]. In *Arabidopsis, TIR1* mRNA level was reduced by overexpression of miR393, which shows late-flowering phenotype through altering the expression of several auxin-responsive genes [68]. The expression of miR160 resistant to AUXIN RESPONSE FACTOR 17 (*ARF17*) shows altered expression of auxin-responsive genes which led to severe developmental defects such as the development of premature inflorescence, sterile and abnormal stamens, decreased petal size and defected root growth in *Arabidopsis* [69]. miRNAs are also involved in limiting the pathogen infection by inhibiting several features of the auxin signaling pathway. For instance, in *Arabidopsis* leaves miR160, and miR393 levels were increased upon infection with *Pseudomonas syringae*, a virulent strain of bacterium [70]. The disruption of gibberellin (GA) biosynthesis pathway and the overexpression of miR159 both delays flowering and reduce fertility, also noticed that by ABA treatment the expression levels of miR159 was upregulated in young seedlings, suggesting that ABA might be involved in inducing the accumulation of miR159 in some tissues [71]. As research progresses, the miRNA/hormone networks are getting more sophisticated, thus requiring more detailed analyses to unravel the genetic regulation beyond hormonal crosstalk in the context of biological processes.

## 3. Regulation of Floral Induction

In plants, flowering is critical for evolution and reproduction. To ensure successful reproduction, plants have to go through distinctive phase changes in their whole life span, from juvenile to adult and adult to reproductive stage. This is attained by controlling the precise expression of important flowering-time genes at both the transcriptional and post-transcriptional levels. miRNAs related to flowering-time act as key positive and negative factors in plant development from vegetative to reproductive phase transition. The involvement of miRNAs and their targets effect and cross-talk with other miRNA-pathways by interacting with biochemical and environmental factors in coordinating the flowering time.

### 3.1. Role of miR156, miR172, and miR390 in Flowering Regulation

miR156 and miR172 are the two key members of the aging pathway and they act together to ensure that the plant produces flowers at an age when they become reproductively competent and have sufficient resources [72]. miR156 indirectly regulates the expression level of miR172, these miRNAs negatively regulate their own sets of target genes in such a way that they retain opposite but related effects in flower regulation [23]. The expression level of miR156 remains high during the early seedling stages and subsequently decreases over time with increasing age of the plant, while the inverse is true for miR172, low expression levels during the juvenile phase and accumulates subsequently during the flower developmental process [57,73]. An evolutionarily conserved role of miR156 in the control of flowering is supported by the fact that overexpression lines of *Arabidopsis*, tobacco, and maize showed delayed flowering phenotypes and extended juvenile phase [74,75,76], whereas overexpression of miR172 accelerates flowering in *Arabidopsis* [74]. Similarly, the highest expression of miR156 is observed during the juvenile phase and consequently declines before floral induction, while the expression of miR172 is low during the juvenile phase and increase subsequently toward floral transition. This expression pattern of these miRNAs is well conserved in *Arabidopsis*, rice and maize [17,74,77]. The crucial roles of miR156 and miR172 in the plant’s lifecycle are evident from the transitional hallmark of flowering.

The eight-member miR156 family (miR156 a–h), miR156a and miR156c plays a leading role in determining the flowering time in *Arabidopsis* [78]. Transgenic plants overexpressing rice *OsmiR156b* and *OsmiR156h* displayed reduced plant stature, reduction in panicle size, and late flowering phenotypes, suggesting that miR156 genes are involved in multiple developmental roles in rice [79]. In another study overexpression of miR156 resulted in prolonged juvenile phase and delayed flowering phenotypes [80], and this effect is more prominent at 16 °C compared to 23 °C, with a higher level of expression detected at low temperature suggest that effect of miR156 overexpression is also influenced by the ambient temperature [81]. The miR156 targets 11 members of the 17 SQUAMOSA PROMOTER BINDING PROTEIN LIKE (*SPL*) family members and their expression levels are negatively correlated, indicating that miR156 primarily controls *SPL* genes at the post-transcriptional level [72]. The *SPL* TF genes sequence contains miRNA-responsive elements, which helps to bind miR156 to regulate their expression [54]. It appears that *SPL* genes function as the plant ‘‘aging genes” and involved in the juvenile-to-adult phase transition [54,79]. Therefore age-dependent decline in the expression level of miR156 is associated with a subsequent increase in *SPL* expression, which promotes floral induction through the *LFY*, *FT* and *MADS-box* gene expression [72]. The phylogenetic analysis shows that orthologues of *SPL-like* genes are present across several species, including 17 members of *Populous trichocapa* [82] and at least 19 members in rice [79], has aided in elucidating the possible roles of *SPLs* in floral regulation. There is prevalent functional redundancy among the *SPL* gene family, and the loss of a single mutant of *SPL* genes often has an inconspicuous effect on plant phenotype [72]. Therefore, *SPL* genes participate in the flowering process can be divided into two groups according to their phylogeny and paralogous relationship. The first group consists of *SPL3*, *SPL4* and *SPL5*, significantly influence the phase change and floral transition [80], and the second group comprises of *SPL9* and *SPL15*, which contribute to the control of flowering time and leaf initiation process [83]. It has been shown that *SPL3* can have direct interaction with intergenic elements and the promoter of floral meristem identity genes (FMIs) *LFY, AP1*, and *FUL* and seems to target the similar FMIs as *SPL9* [84]. In *Arabidopsis* response of *SPL3/4/5* toward floral induction is more prominent and strongly expressed in comparison to *SPL9/15* [85]. In the early stages of vegetative development, *SPL3/4/5* expression is affected by day length, being higher in LD and lower in SD conditions [86]. However, at lateral vegetative stages, *SPL3/4/5* expression is independent of photoperiod and correlates with floral induction [85]. This effect of *SPL3/4/5* to day length is accompanied by the photoperiod pathway genes *CO* and *FT* and irrespective of miR156, as expression levels of miR156 are unchanged with regard to different photoperiod [86]. However, more recent study shows that knocking out all three *SPL3/4/5* using CRISPR does not affect the vegetative phase change or floral induction in *Arabidopsis*, but promotes the floral meristem identity transition [87]. The double mutant of paralogous genes, with *spl9 spl15* displaying similar distinct phenotypes to that of plants overexpressing miR156 [88]. Conversely, transgenic lines expressing miR156-resistant *rSPL9* or *rSPL15* lead to the production of adult leaves and accelerated flowering [80]. It was revealed that early flowering phenotypes resulted due to the induction of miR172 expression by *SPL9* [15]. In addition to miR172, *SPL9* has been shown to directly regulate the expression of *AP1*, *FUL*, *AGL24* and *SOC1* through binding to their respective promoters (Figure 2) [54].

As the plants grow older, the level of miR156 declines so does the repression of flowering. The effectors which cause a decline in miR156 expression level over age are not well understood. As plants get older sugar accumulates in shoot meristem, which serves as a signal for the availability of carbohydrates (glucose and sucrose) in plants, selectively regulates the expression of miR156 targets [78,89]. An increase in sugar assimilation reduces the expression of miR156 and conversely decline in sugar deprivation was shown to increase miR156 expression with a subsequent increase or decrease in expression levels of *SPL*, respectively. The floral induction in plants is greatly associated with increased sugar accumulation, as loss-of-function mutants in carbohydrate metabolism displayed late-flowering phenotypes [90]. In *Arabidopsis*, under LD conditions assimilation of sucrose induces *FT* expression for flowering [91]. A study revealed that glucose-sensing enzyme Hexokinase 1 (*HXK1*) is required for the upregulation of miR156A and miR156C expression in low sugar conditions [89]. This sugar-mediated expression of miR156 is conserved in different plant species [78]. Accumulation of trehalose-6-phosphate (*T6P*) promotes flowering and established a link between miR156 expression and plant carbohydrates status [92]. The enzyme trehalose-6-phosphate synthase 1 (TPS1) converts UDP-glucose and glucose-6-PO_4_ into T6P which function as a signal for carbohydrates availability in plants [92]. In *Arabidopsis*, transgenic lines with loss of *TPS1* expression showed a reduced level of *T6P* and caused extremely late flowering, indicating that *TSP1* down-regulates miR156 expression with an increasing level of SPL transcripts [92].

The miR172 family is encoded by five loci miR172a-e acts downstream of miR156 and has the opposite effect on flowering time [15]. The expression levels of miR172a-c increases as plants get into a reproductive developmental phase, while the expression of miR172d-e is age-dependent and extremely low during this phase [57]. In *Arabidopsis*, transgenic plants overexpressing miR172 resulted in accelerated flowering under both LD and SD conditions [52]. The level of miR172 gradually increases during vegetative to reproductive phase transition, indicating the crucial role of miR172 in controlling the plant developmental changes [86]. The miR172 promoter contains several copies of *SPL* binding elements, which act as a transcriptional activator. Transgenic lines overexpressing *SPL9* and chromatin immunoprecipitation experiments showed an increased level of miR172. The temporally opposite expression pattern of miR156 and miR172 is thus a direct consequence of the decline in miR156 level and increased expression of SPL genes [52]. In *Arabidopsis* miR172 negatively regulate the expression of *AP2* and five AP2-like genes including TARGET OF EAT 1 (*TOE1*), *TOE2*, *TOE3*, SCHNARCHZAPFEN (*SNZ*) and SCHLAFMUTZE (*SMZ*) [36,72]. MiR172 controls its target genes primarily by translation inhabitation; however, transcript cleavage has also been observed [93]. All the *AP2-like* genes act as a floral repressor and delays flowering through inhibiting the expression of floral integrator genes *FT*, *FUL*, *LFY* and *SOC1* [94]. The miR172 overexpression or its target genes (excluding TOE3) demonstrate both early and late flowering phenotypes under LD and SD conditions, respectively [57]. The *toe1 toe2* double mutant flowered earlier but still flowered late in comparison to the plants overexpressing miR172. These findings led to the assumption that there could be additional factors acting together with *TOE1* and *TOE2* in flowering time control. Later this was proved to be true with the discovery of *SMZ* and *SNZ* and the *toe1–toe2–smz–snz* quadruple mutant exhibited the early flowering phenotypes than the double mutant *toe1 toe2*. However, the quadruple mutant still flowered late compared to the plants overexpressing miR172, suggesting that there might be other redundant factors playing additional roles [95]. Eventually, when *AP2* was identified as a flowering repressor, the loss-of-function hextuple mutant of all the six miR172 target genes was developed, which could phenocopy the early flowering feature of miR172 overexpression [96]. AP2-like protein levels are high during the juvenile stage and declines as the level of miR172 increases with an increase in plant age; hence, indicating the flowering repression as the plant moves toward maturity [15]. Interestingly, *AP2* act in a feedback loop by which miR156 is up-regulated and miR172 is down-regulated [80], further complexity arises by AP2-like proteins regulating the expression levels of other *AP2-*like genes, an additional layer of the feedback loop in order to fine-tune the flowering mechanism [96]. 

miR172 expression is also regulated by photoperiod flowering pathways and ambient temperature [72,97]. GIGANTEA *(GI)*, a clock-associated gene, known to contribute to photoperiodic flowering and regulates *CO* at the transcriptional level in inductive light conditions [98]. The miR172 showed reduced expression in the *gi* mutant; however, miR172 abundance increased significantly in LD photoperiod in both *gi* mutants and wild type, compared to plants grown under SD conditions. The *gi* mutant showed reduced expression of miR172 despite the increase in primary MIR172 (priMIR172) transcript levels, which indicates a *GI* role in miR172 processing rather than its transcription [57]. This could be due to the fact that the miRNA processing enzymes including SE and DCL1 were also observed to be declined in the *gi* mutants [57]. Similar to miR156, the expression of miR172 is influenced by the ambient temperature, miR172 expression was higher at 23 °C, compared to 16 °C, presenting a quite opposite pattern to that of miR156 [97]. The ambient temperature-dependent regulation of miR172 is influenced by a flowering repressor SVP protein, which down-regulates the miR172 expression. In the *svp* mutant, expression levels of miR172 are elevated compared to wild-type plants at both 16 °C and 23 °C, with a simultaneous decline in the expression of miR172 targets [97]. However, miR172 showed decreased expression at 16 °C in the *svp-41* mutant, as compared to wild-type plants at 23 °C, suggesting that upregulation of SVP components are dependent on low ambient temperature and results in decline of miR172 expression, with a temperature-dependent increase in the level of miR156 contribute to this phenomenon [72,99]. 

The miR390 family participates in many different development processes such as root development, leaf morphogenesis and influences the flowering process directly or indirectly. miR390 affects flowering time through prolonging the juvenile phase, as a result, delays the acquisition of the competence to flower [67,100]. The effect of miR390 on flowering time is not only because of targeting protein-coding mRNAs but also trigging the production of ta-siRNAs from the *TAS3* locus, which in return represses the mRNA levels of TFs *ARF3* and *ARF4* [101]. The activity of these TFs promotes the vegetative-to-reproductive phase transition, the activity of miR390 inhibit flowering through extending the vegetative phase. In *Arabidopsis*, ta-siRNA-defective mutants showed accelerated juvenile to adult phase transition, due to an increased level of *ARF3* and *ARF4* [67]. The ta-siRNA-insensitive *ARF3* (*ARF3*: *ARF3*mut) expression in transgenic plants showed the same phenotype [67]. This indicates that miR390 represses flowering by inhibiting the activity of *ARF3* and *ARF4*, which results in prolonging the juvenile phase [101]. 

Increased activity of *ARF3* and *ARF4* may affect the miR156-regulated SPL genes by affecting the juvenile to adult transition [101], as *AP2* represses the *ARF3* expression by directly binding to its promoter [96]. *ARF3/4* may regulate the expression of *SPL3*, directly or indirectly [101]. Therefore, it establishes a link between the miR156/miR172 and miR390 by interacting in a feedback loop to regulate the juvenile-to-adult transition in certain aspects of the aging pathway.

### 3.2. Role of the miR159, miR169, miR172 and miR399 in Flowering Regulation

The three-membered miR159 family affects flowering time by its functions in GA-mediated floral regulation, and also participate in a network together with other miRNAs such as miR167 and miR319, which play an important role in controlling the floral organ development. However, the miR159 function in controlling flowering time is not clear due to conflicting evidence. The miR159 is a homeostatic modulator of GAMYB or GAMYB-type genes that encode MYB domain TFs have been implicated in the GA flowering pathway. In response to GA signals, *MYB* TFs binds to the *LFY* promoter elements to directly activate its expression, which accelerates flowering in non-inductive SD conditions [72,102].

Recent studies have demonstrated that miR159 overexpression causes a decline in the activity of *LFY* and *MYB33* and showed late flowering phenotypes in an ornamental plant Gloxinia (*Sinningia speciosa*) and *Arabidopsis* [71,103]. In rice, Overexpression of miR159 also resulted in delayed flowering phenotypes [104], and some lines showed more significant delayed flowering than others, this could be due to the difference in the expression levels of transgenes, transgenic lines having higher expression level cause more significant delay. However, *Arabidopsis* lines overexpressing miR159 did not display any delay in flowering time, the reason for that could be the expression of miR159 in transgenic *Arabidopsis* was not high enough that could lead to a decline in GAMYB level and delay flowering.

The DELLA protein represses the GA response that leads to a decline in the expression of both miR159 and its target *GAMYAB*. GA treatment degrades DEELA protein and increases the levels of miR159 and *GAMYB* TFs, which bind to the *LFY* promoter through GA-responsive *cis*-elements to activate its transcription and promote flowering [71,105]. As miR159 promoter contains several putative GARE-like sites [71], suggesting that *GAMYB* factors may also involve in enhancing miR156 expression in a feedback loop by down-regulating the expression of *GAMYB* and may act as a level of homeostatic regulation of GA response. Intriguingly, even as overexpression of miR159 cause a decline in *GAMYAB* levels and delayed flowering phenotypes, several recent studies supported by the fact that eliminating the inhibition of *GAMYAB* expression either in miR159 mutants or by the overexpression of a miR159-resistant *MYB33* gene did not result in early flowering [71,106]. This is based on the fact that DELLA proteins also act to suppress the expression of the *GAMYB* gene and perhaps will be able to maintain comparatively normal *GAMYAB* levels. 

The miR169 is one of the most abundant miRNA families with 14 members (miR169a–n) in *Arabidopsis.* miR169 is known to control flowering time in response to various stresses. Most of the miR169 family members in *Arabidopsis*, soybean, and maize are upregulated in response to abiotic stress. The main target of miR169 is the NUCLEAR FACTORY, SUBUNIT A (*NF-YA*) TF gene family, which is associated with transcriptional regulation of multiple genes [106,107]. The miR169d overexpression in *Arabidopsis* exhibits early flowering, and in contrast, overexpression of the *rNF-YA2* accounts for late flowering [108], indicating that miR169 represses the expression of *NF-YA2* and regulates stress-induced flowering. Plants overexpressing miR169d with a reduced level of *NF-YA2* showed a decline in the expression of FLOWERING LOCUS C (FLC), whereas *FT* and *LFY* target genes of FLC showed significantly higher expression levels. The *NF-YA2* regulates FLC by binding to its promoter to induce its expression, thus increased expression of *LFY* and *FT* promotes flowering in miR169d overexpressing plants. Conversely, expression levels of FLC were significantly increased in plants overexpressing miR169d-resistant *NF-YA2*, which result in the reduction of *FT* and *LFY* expression levels and exhibited late flowering phenotypes [108]. These results suggest that the miR169 regulatory pathway functions independently to that of the miR156/miR172 pathway and plays a critical role in promoting early flowering through the abiotic stress response.

The miR399 is a perfect example of miRNAs involved in both control of flowering time and abiotic stress response. MiR399 family consists of six family members (miR399a–f). It was identified that miR399 plays a crucial role in regulating the phosphate homeostasis, by downregulating the expression levels of the PHOSPHATE 2 (*PHO2*) gene [109]. *PHO2* encodes E2 ubiquitin-conjugation enzyme that regulates the proteins involved in phosphate uptake [109]. Phosphate is one of the key essential elements involved in maintaining different cellular processes and its depletion causes severe damaging effects on plant development [110]. The activity of miR399 is up-regulated in phosphate starvation conditions, which helps to improve phosphate uptake in roots; on the other hand, miR399 activity is downregulated to prevent phosphate toxicity in high phosphate conditions. Therefore, its activity is tightly regulated to avoid unnecessary phosphate accumulation which can cause tissue necrosis [110,111]. More recently, it was shown that accumulation of miR399 is influenced by ambient temperature, as the levels of miR399 were higher and plants exhibited early flowering phenotypes grown at 23 °C; however, no change in flowering time noticed at 16 °C [109]. Kim et al. (2011) demonstrated that both miR399 overexpression and loss-of-function *PHO2* mutants showed early flowering phenotypes when grown at 23 °C, while unaltered flowering time observed of these mutants when exposed to lower ambient temperature at 16 °C, this could be due to the fact that increased level of the floral pathway integrator TSF expression at 23 °C may lead to accelerated flowering of these plants [109]. Thus, miR399 is one of another example of miRNAs participate in flowering time control and abiotic stress response.

The role of miR156-SPL and miR172-AP2 nodes of the aging pathway, along with miR319-TCP, miR159-MYB and miR399-PHO2 nodes have considerably increased our knowledge about the mechanism of miRNAs underlying floral transition (Figure 2). However, there are still a lot remained to be undiscovered. The role of miR156 in prolonging the vegetative phase and delaying flowering, as well as the nutrient-dependent signals mediating its age-related decline is still not fully understood. The factors involved in regulating the expression of miR319 are yet to be further explored. In general, the impact of miRNA-target interactions, and the complex interplay of these miRNAs involved in various flowering pathways are the areas of future research.

## 4. Role of miRNAs in Inflorescence Development

### 4.1. Shoot Apical mEristem Development

Endogenous and exogenous cues are integrated through a complex genetic network consisting of numerous overlapping pathways in order to activate the expression of key flowering genes in the SAM (shoot apical meristem) [72]. When the expression of flowering genes reaches a certain level, the SAM switches from a vegetative meristem to a floral meristem and flowers are produced. One member of the AGO family in *Arabidopsis* has demonstrated that *AGO10* is a key regulator of proper SAM maintenance [112,113]. Experiments in transgenic plants showed that *AGO10* inhibits miR165/166 expression, and these two miRNAs differ in only one nucleotide in the mature miRNA sequence [112]. The interaction between *AGO10* and miR165/166 can specifically release the class III HOMEODOMAINLEUCINE ZIPPER *(HD-ZIP III)* gene expression and maintain proper SAM development; similar findings were reported in peach [94]. A microarray study in soybean found 31 miRNAs and six putative legume novel miRNAs expressed in the SAM, suggesting that they play different crucial roles in mediating SAM development. Another conserved miRNA that highly expresses in SAM is miR159, it regulates the expression of *MYB33* and *MYB65*, and is found throughout the SAM. This observation was consistent with the expression level of miR159 in *Arabidopsis* [114]. This indicates that a complex network mediated by many different miRNAs are regulating the SAM development.

### 4.2. Flower Patterning

Once the plants have completed the floral transition phase, the floral meristem identity genes induce the expression of different floral organ identity genes regulated by miRNAs, and thus directing the conversion of floral meristems into floral organ primordia. The floral organ development depends on the effect of floral organ identity genes and appears in successive whorls. In *Arabidopsis*, it consists of a whorl of four-petal primordia, a whorl of six stamens, and two carpel primordia. The formation of these distinct whorls requires us to establish boundaries to limit the expression of genes that controls the floral patterning as per the ABC model [115]. The activation of the ABC model genes specifies a floral organ identity. *AP2* is a class A gene known for its role in sepal and petal development by interacting with other A or C class homeotic genes and acts mutually with the C class homeotic gene AGMAOUS (*AG*) [116]. The accumulation of miR172 takes place in the center of flower primordia, which restricts *AP2* expression to the two outer whorls of the floral meristem, and determines the boundaries between stamens and petals [117]. It was shown by in situ hybradyzation experiments that the domains of expression of miR172 in the inner floral whorls and the *AP2* in the outer floral are largely complementary [117,118]. On the other hand, at some time points, an overlap in the expression domains of *AP2* has been observed suggesting that additional factors other than miR172 could be involved in regulating the extent of *AP2* expression [118]. Grigorova et al. (2011) demonstrated that LEUNIG (*LUG*) a transcriptional regulator, act upstream of miR172 and directly represses its expression in sepals and *AP2* presence is required for this repression [119]. These findings suggest that a negative feedback loop in which *AP2* downregulates the expression of miR172 is crucial for the accurate floral organ boundary formation during flower development [96]. It is also supposed that *AP2* is involved in conscripting the *LUG* transcriptional repressor complex to negatively regulate the expression of miR172 in the outer floral whorls, which result in maintaining its own expression in those whorls [119]. Overexpression of miR172 has been shown to affect floral patterning in several cereal crops such as barley and rice, where it targets the *AP2-like* TFs CLEISTOGAMY1 (*cly1*) in barley, and Oryza sativa INTERMINATE SPIKELET1 (*osIDS1*) and SUPERNUMARY BRACT (*SNB*) in rice, to control the lodicule development [120,121]. 

In *Arabidopsis,* miR164 regulates the expression of NAC domain family members of TFs, including *CUC1* and *CUC2* genes [44,122]. The regulation of these TFs by miR164 is necessary for the accurate floral organ boundary formation during the establishment of the floral meristem. Reduction in the activity of *CUC1* and *CUC2* genes displayed fused sepals and reduction in the number of formed petals, while loss-of-function mutants of miR164c showed the formation of extra petals [123]. In tomato, miR164 targets GOBLET (*GOB*), *NAC1* and *NAM2* genes, which are required for setting boundaries between SAM and leaf primordia. The tomato plants overexpressing miR164 showed defective development of floral organs and leaf [124]. Besides, interestingly, it was found that miR164 is expressed abundantly in some fruits including orange [125], prickly pear fruit [126], tomato [127] and grapes [128]. The expression of miR164 was remarkably higher in developed orange fruits, compared to other organs [125]. Furthermore, miR164 accumulation was at the highest level during the fruit ripening process [125].

The miR169 is involved in repressing the expression of class-C genes. In Antirrhinum and Petunia, miR169 regulates the several members of the *NF-YA* TF gene family, that are supposed to contribute to flower development [107]. The miR169 encodes BLIND (*BL*) in Petunia and FISTULATA (*FIS*) in Antirrhinum, and these encoding genes activate the expression of *NF-YA* TF genes, which restrict the activity of class-C genes to the inner two whorls through post-transcriptional repression to specify the reproductive floral organs during the process of flower development [129]. The *FIS* and *BL* loss-of-function mutants displayed stamenoid petals in their second whorls, suggesting that the aberrant behavior of C-gene in the second whorl is due to the reduced activity of miR169 [129]. However, such functions of miR169 in reproductive organ development do not seem to be the case in *Arabidopsis*, where restricted C-gene activity has not been observed [129].

### 4.3. Flower Development 

Many different miRNAs play pivotal roles in critical developmental processes such as flower patterning, where floral organs become complex structures. Due to their distinct overlapping functions, miR319 and miR159 have paramount importance in the flower development process through regulating the *TCP* and *MYB* TFs, respectively. Both TFs interact with each other to regulate the miR167 expression thus, forming a regulatory circuit that facilitates the flowering development. miR319 and miR159 have 17 nucleotides identical between them and they evolved from a common ancestor [130]. Despite the fact these miRNAs do not cross-regulate each other’s targets, miR319 can bind to the *MYB* transcripts, whereas miR159 is not capable of binding *TCP* transcripts, indicating that miR319 has a limited spatiotemporal expression pattern, in comparison with miR159.

The gene family of miR319 is made up of three members miR319a-c and targets a large subset of *TCP* TF genes including *TCP2*, *TCP3*, *TCP4*, *TCP10* and *TCP24*, these TFs have been well described to play major roles in different developmental processes such as flower production, and gametophyte and leaf development [131]. Overexpression of miR319 in *Arabidopsis* positively regulates the cotyledon and leaf development, at the same time causes stamen and male sterility defects [132]. Whereas miR319 loss-of-function mutant showed defects in stamen and petal formation, including short and narrower petals and defective anther development [133]. 

As mentioned earlier, miR159 downregulates the expression of some *GAMYB*-related genes, in addition, it also regulates the *LFY* transcript levels and has a serious impact on anther formation and control of flowering time [71]. Transgenic *Arabidopsis* plants overexpressing miR159 have a reduced level of *MYB33* and displayed pleiotropic developmental defects which include stunted anther development, reduced fertility, and small siliques, consistent with what has been observed in plants overexpressing miR319 [71,134]. In rice, miR159 also restricts the *OsGAMYB* expression to anthers [104] and a loss-of-function *OsGAMYB* mutant displayed defective anther and pollen formation [135]. Remarkably, miR159 accumulation is elevated by gibberellin (GA) application, while lower levels of miR159 were observed in GA-deficient mutants [114].

The reduced activity of miR319 and miR159 through the expression of target mimics display developmental defects in petals, sepals and anthers formation, similar defects have been noticed in *arf6 arf8* double mutant [101]. It has long been recognized that *ARF6* and *ARF8* regulate the levels of auxin homeostatic genes by specifically binding to the auxin-responsive element (AuxREs) present in the auxin-responsive gene promoters [136]. In *Arabidopsis*, miR167 regulates the expression patterns of *ARF6* and *ARF8* in floral organs [137], and increased level of miR167 or reduced expressions of *ARF6/8* leads to similar floral phenotypes, which includes the formation of defective floral organ and reduced fertility [138]. The *MYB33* and *TCP4* independently upregulate the expression of miR167, which are themselves controlled by miR159 and miR319, respectively. Therefore, this overlapping function of miR159 and miR319 could be attributed to their targets (*MYB* and *TCP*) along with miR167 target genes *ARF6* and *ARF8* (Figure 3) [101].

miR160 downregulates the expression of *ARF10, ARF16* and *ARF17* genes [139]. A loss-of-function in the floral organs in carpel *(foc*) mutants, there is a transposon insertion in the 3′ regulatory region of miR160, owing to reduce expression of miR160 in flowers. Therefore, the expression levels of *ARF10, ARF16* and *ARF17* are increased in *foc* mutants and exhibit some defects of flower formation, including the development of irregular flower shape, reduced fertility and aberrant seed production [139]. Another study established that the expression level of miR160a is also found to be controlled by auxin, and positively regulates its expression by binding to auxin-responsive element present in the promoter region of miR160 gene [139]. 

### 4.4. Sex Determination in Flowers

Maize is monoecious, with two types of inflorescences, the male tassel, and the female ears. Flowers are produced as bisexual, but later undergo stamen arrest within the ear and pistil abortion in the tassel. The analysis of mutants has shown to affect sex determination in maize. In tassels the tasselseed4 (*ts4*) and Tasselseed6 (*Ts6*) mutants, pistel often fails to abort and male floral organs could not able to develop, resulting in the development of pistils in the tassel rather than the stamens. Cloning of the ts4 gene revealed to be a member of the miR172 gene family, annotated as *zmaMIR172e*. Sequencing analysis of *zma-MIR172e* from three *ts4* mutants confirmed that mutations in *zmaMIR172e* were responsible for *ts4* phenotypes. Mutations in the *AP2*-like gene INDETERMINATE SPIKELET1 (*IDS1*) targeted by zma-miR172e, results in partial suppression of ts4 phenotypes [140]. Cloning of *Ts6* showed that it is a mutated form of *IDS1* gene, sequence comparison of *TS6* and *IDS1* presented that *Ts6* mutations alter the miR172 binding site thus preventing regulation by zma-miR172e which lead to similar phenotypes of both *ts4* and *ts6* mutants, that include failure of carpel abortion and floral meristem indeterminacy [140]. 

Interestingly, as Chuck et al. point out, the wheat domestication gene Q encodes an *AP2*-like TF gene orthologous to *IDS1*. A single nucleotide (C-to-T) polymorphism at the 3’ end of the miR172 binding site gave rise to the Q allele which is found in the cultivated wheat [140] and shown to have higher expression levels compared to the recessive q allele [141]. It can be supposed that other mutations similar to Q, ts4, and ts6 may result in altered regulation of miRNAs involved in inflorescence development could have a role in the domestication of other crop species.

The development of male sterility systems in different crops is probably the most apparent economic benefit [142]. The factors involved in regulating the expression of miR319 are yet to be further explored. The other potential miRNA approaches could modify flower appearance/shape, and possibly even control the sex of the flower in different plant species [14]. Until now, a limited amount of genetic variations in plant miRNAs and their regulatory function in gene product activity have been functionally characterized. It is necessary to functionally characterize the genetic variation in other miRNAs and their targets and provide trait enhancing alleles for crop improvement.

## 5. Biotechnological Applications for miRNAs in Controlling Flowering Processes

miRNAs are involved in nearly every aspect of plant growth and development. Different methods can be used to alter the expression level of endogenous miRNAs, or the use of amiRNAs (artificial miRNAs), provides the opportunity to regulate key TFs and thus whole gene regulatory network [143]. The modification of plant miRNAs has the potential to improve economically essential processes such as plant development, flower regulation and increased yield, which can contribute to agricultural and horticultural productivity in order to develop superior cultivars.

### 5.1. Flower Regulation 

Flowering is an important development process and it is directly linked to the production potential. Among the factors affecting crop yield, flowering time and instability of blossoming are the most challenging issues, thus flowering at an appropriate time is vital for crop yield and marketing purposes. On the other hand, the development of new varieties particularly in tree species, with characteristics of having a shorter juvenile phase and ability to induce early flowering will result in more yield and beneficial for the farmers.

It is possible to enhance the activity of miRNAs by overexpressing the MIR genes or amiRNAs (artificial miRNAs) using the tissue-specific or constitutive promoters. Conversely, the activity of specific miRNAs can be sequestered through an expression of target mimics [143,144]. Schwab et al. engineered artificial miRNAs to regulate the expression of the *Arabidopsis FT* gene (*amiR-FT),* in which the expression of the *FT* gene was down-regulated to get delayed flowering phenotypes in transgenic plants. The levels of *FT* mRNA were lower than those of qRT-PCR detection levels in *amiR**-ft* expressing plants and the extent of delayed flowering in *amiR-ft* line was similar to that of ft null mutants [143]. A similar inhabitation of flowering was observed when *amir-FT* was expressed under the control of *SUC2* promoter, which is known to express in phloem companion cells [145]. The *FT* gene acts with a partially redundant and closely related paralog *TSF* (TWIN SISTER OF FT), and *ft tsf* double mutant even displayed very late flowering than the *ft* mutant. The amiRNAs can be used to downregulate more than one target if target genes have enough homologous sequence for an amiRNA binding site [143]. Mathieu et al. designed an amiRNA that could target a specific sequence in both *FT* and *TSF* genes. The ami-FT/TSF construct in *Arabidopsis* plants displayed late-flowering phenotypes as does the *ft tsf* double mutant [145]. Li et al., 2013 displayed that it is possible to control both early and delayed flowering production in ornamental Gloxinia (*Sinningia speciosa*) by either over- or underexpression of miR159, which is a negative regulator of Gloxinia homologues *GAMYB* and *LFY.* Transgenic lines overexpressing miR159 were delayed in the onset of flower production, while those lines in which activity of miR159 was attenuated resulted in early flowering phenotypes [103]. Few transgenic lines in which levels of miR159 were too high and lines where miR159 was strongly repressed and high expression level of *LFY*, showed the conversion of some petals into sepals. Manipulating the level of miRNAs using inducible expression systems would enable the control of flowering time in a precise manner.

The understanding of miRNA-based target regulation would help to design and enhance important agronomical traits for crop improvement. In most cases, the overexpression or knockout of miRNA targets usually exhibits almost similar phenotypes, since many miRNAs regulate their target genes only in certain specific types of cells, and exclusively under certain conditions. However, the effects of many essential miRNAs in living cells may not be visible due to the insufficient level of expression. In such circumstances, it would be crucial to perform quantitative analysis of the natural plant miRNAs before designing the artificial miRNAs for desirable agronomical traits [146,147]. As the degree of down-regulation of different target genes by an amiRNA can also vary, and the level of variation is not correlated with the degree of complementarity to the target or the level of endogenous expression of the target gene [143]. Although this artificial miRNA method is already employed as a useful tool, the evaluation and differentiation of mRNA cleavage and translational inhibition still requires more clarification for better application.

### 5.2. Modification of miRNAs or Their Targets via CRISPR–Cas9 Technology

In recent years, several methods have been developed and modified for the study of miRNAs loss-of-function, including sponges and antisense inhibitors. However, the stability and specificity of these techniques are not highly accurate. Recently CRISPER-Cas9 (clustered regularly interspaced short palindromic repeat-CRISPER associated protein 9) is emerging as a breakthrough in the technology for genome editing tool for the field of plant biology in last decade [148]. With the help of specifically designed RNAs (sgRNAs), Cas9 can cut DNA sequence from almost any desired location in the genome to generate double-strand break (DSB), which are then repaired via non-homologous end joining (NHEJ) [149]. Ultimately, the frameshift occurring in the coding region brings expressional changes or gene silencing [149]. 

The CRISPR-Cas9 system has been used successfully to edit multiple genes and also to knock out protein-coding genes in model plants like *Nicotiana benthamina* and *Arabidopsis* and some essential crops such as maize, rice, wheat, sweet orange and tomato [150,151,152,153]. In rice and *Arabidopsis* plants containing a CRISPR/Cas9 transgene has been reported with detectable mutation as high as 90% [154,155]. This has modernized the practices of traditional breeding by creating an efficient genetic variant of crop plants successfully. However, all these studies impetus on the protein-coding genes, evidence regarding the targeting of non-coding RNA genes including miRNA and lncRNA genes, is less reported. It can be supposed that cas9 should be an innovative approach in modulating the expression of miRNAs. One of the most challenging issues in the target modification is an ineffective method of delivery system to plant cells. To encounter this issue, a new effective vector system developed by using Geminiviruses [156]. It was established using the model plant *Arabidopsis* that Geminiviral-sequence is helpful in homologous repairing of DSB [156]. As a number of miRNAs operated from multiple loci, so the manipulation of certain loci might not outcome as a proposed functional change. Furthermore, practicing CRISPER-Cas9 to knock out or knock down miRNA genes, particularly when located in the intron of protein-coding genes is assumed to be challenging, because there is every chance of affecting nearby or host genes in the vicinity [157]. Nevertheless, CRISPER-Cas9 has the potential to knock down or knock out miRNAs efficiently by targeting miRNA genes at multiple sites such as hairpin and promoter [157]. 

Recently successful application of CRISPR–Cas9 used in rice for gene/base replacement has broadened the scope of crop improvement. In recent past, as several important agricultural traits in crops are obtained by point mutation in miRNAs or their targets [158,159], application of CRISPR–Cas9 can be used to alter or disrupt the promoter enhancer elements such as transcriptional start sites (GC-box or TATA) and other trans-acting factors (ABRE and DRE motifs) to retain precise miRNA expression that can assist to regulate flowering in higher woody plants [160]. The combinations of a nicked Cas9 with the Asp10Ala mutation and a cytidine deaminase enzyme, which retains the capability to be programmed by a guide RNA without the induction of double-strand DNA breaks, and facilitates the conversation of uridine to cytidine, thus precisely resulting in G to A or C to T base replacement within the window of approximately five nucleotides [161,162]. Another method of targeting promoters can be performed with the help of a catalytically inactive Cas9 in association with sgRNA for the accurate intervention of transitional machinery [163]. If CRISPER-cas can somehow be designed in a way that it can target the 3′ or 5′ arm of mature miRNA. The CRISPER-cas technology can be used to generating mutant miRNA binding sites in target genes, which also helps to verify miRNA targeting [164].

CRISPR-Cas9 provides an effective and efficient alternative for the functional characterization of miRNAs. Once the CRISPR-Cas9 mediated knock-in or knockout mutation is created, the high-throughput or next-generation sequencing technology can be used to profile the expression of miRNAs and their targets in biological pathways. These technologies provide new approaches for the functional characterization of miRNAs and effective way of dissecting flowering mechanisms in higher woody plants. 

## Figures and Tables

**Figure 1 genes-11-00319-f001:**
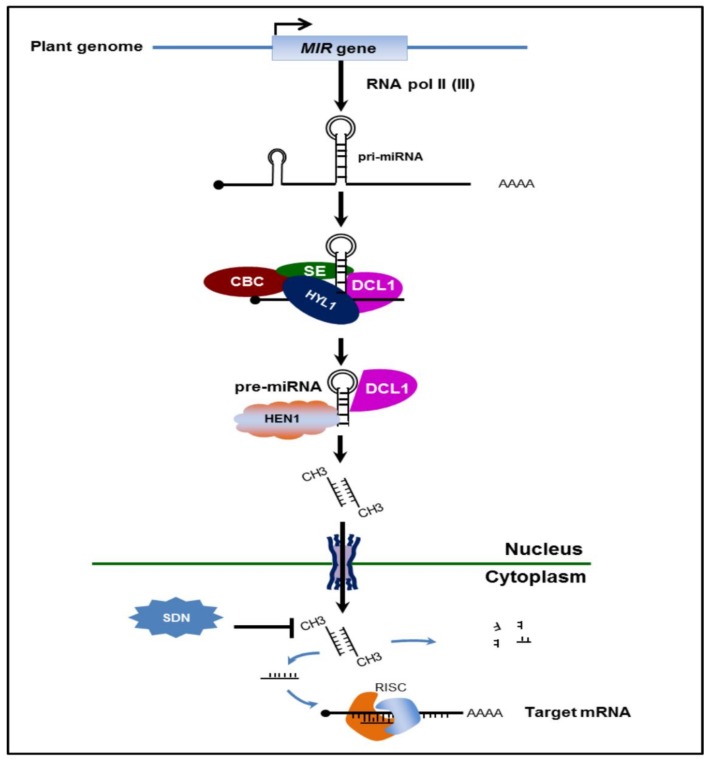
Scheme of plant microRNA biogenesis. Plant microRNA are transcribed by RNA poly II from the transcripts located in intergenic regions. Pri-miRNA was stabilized by DDL, then binds with Serrate (SE), Hyponasty Leaves (HYL1), and cap-binding complex (CBC), and processed by DCL1 into pre-miRNA. The pre-miRNA duplex is methylated by Hua Enhancer 1 (HEN1) and transported from the nucleus to the cytoplasm by HASTY. The guide miRNA strand binds AGO1 and is loaded into the RISC complex to carry out the silencing process.

**Figure 2 genes-11-00319-f002:**
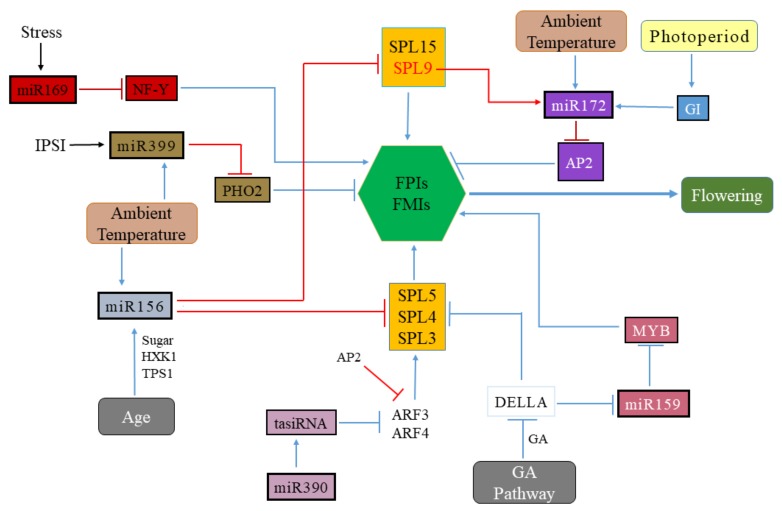
Analysis of different miRNAs families and their target genes implicated in the regulation of flower induction. The accumulation of miR156 subsequently declines with an increase in plant age, other factors such as sugar (HXK1) accumulation and the ambient temperature gradually repress miRNA156 expression, resulting in rising levels of SPL family genes. MiR172 expression is positively regulated by GI, that leads to the down-regulation of AP2-type floral repressors, miR172 is also regulated by ambient temperature. The miR159 is regulated by the GA pathway and in turn, miR159 regulates the MYB transcription factors. The DELLA protein represses the expression of miR159 and the SPL3/4/5 genes. The miR390-ARF3/4 module regulates floral transition indirectly by inducing the production of ta-siRNAs from the TAS3 locus, through putative control over SPL gene expression. The ambient temperature-induced miR399-PHO2 module and stress-induced miR169-NF-Y module regulate floral induction by regulating the expression of floral meristem identity genes. Overall, miRNAs can either promote or inhibit the expression of the FMI (floral meristem identity genes) and/or FPI (floral pathway integrator), thereby promoting or delaying the onset of flowering, respectively. *AP2, APETALA2; ARF, AUXIN RESPONSE FACTOR; HXK1, hexokinase; GI, GIGANTEA; NF-Y, NUCLEAR FACTOR Y; GA, gibberellic acid; SPL, SQUAMOSA PROMOTER-BINDING PROTEIN-LIKE; PHO2, PHOSPHATE 2; ta-siRNA, trans-acting siRNA; TPS1, TREHALOSE-6-PHOSPHATE SYNTHASE1; PHO2, PHOSPHATE 2*.

**Figure 3 genes-11-00319-f003:**
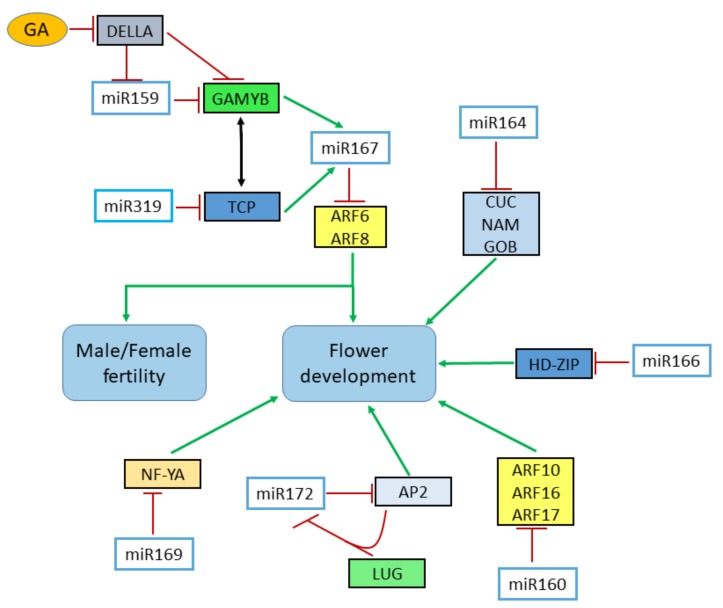
Analysis of different miRNA families and their target genes involved in genes in setting boundaries, flower development, and male/female fertility. MiR319 and miR159 and their target TCP proteins and GAMYB interact with miR167 and in return down-regulates the level of ARF6 and ARF8. GA treatment causes the down-regulation of both miR159 and GAMYB transcription factors through the degradation of DELLA proteins. MiR160 represses the expression of ARF genes which ultimately contribute to flower development. *ARF, AUXIN RESPONSE FACTOR; GA, Gibberellic acid; CUC, CUP-SHAPED COTYLEDON; AP2, APETELA 2; GOB, GOBLET; LUG, LEUNIG; LFY, LEAFY; NAM, NO APICAL MERISTEM; NF-Y, NUCLEAR FACTOR Y.*

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
