# Peer review of "The Critical Role of miRNAs in Regulation of Flowering Time and Flower Development"

_genes, 2020, doi:10.3390/genes11030319_

Round 1

Reviewer 1 Report

The manuscript "The critical role of miRNAs in flower regulation and development" presented the broad roles of plant miRNAs in controlling flowering and other organs development. Overall, this work covered most recent research information of plant miRNA. However,this work is not well organized and many grama errors. And in addition, it's very confused that different format or colours of letters could be found through whole the manuscript. Also the format of reference is inaccurate.
Some errors could be easily found, Such as:

1. Abstract: "require" should be "requires"; in last sentence "are" should be ruled out;
2. Introduction: "undergo" should be "undergoes", "play role" to "play a role" in first sentence; "MicroRNA" should be "MicroRNAs" in second MicroRNA;
3. "Mirna" to "MiRNA" or "miRNA"; Letters for plant species should be in italic.......

Reviewer 2 Report

The authors reviewed the regulatory functions of miRNAs on flowering and flower development. This review covered the mechanism of miRNA regulation, the role of miRNA in many aspects of plant development but with a focused on the role of miRNA in floral induction and development.

Major comments:

(1) There are already some reviews about the role of miRNAs in plant flowering (see doi links below). In the introduction, the authors should mentioned existing reviews on the same topic and how this review have expended and advanced from what has already been reviewed. In the introduction and abstract, the author should justify why this review is worth to publish given there are already some similar review papers.

https://doi.org/10.1016/j.molp.2014.12.018

https://doi.org/10.1093/jxb/ert453

(2) In the end of section 3 and 4, could the author add a paragraph to summarize what are the missing puzzles in these topics, what are still unknown and worth to explore. I understand the author mentioned some throughout the manuscript but would be better to make a summary in the end.

(3) In section 5.1, the authors discussed engineering amiRNA to control flowering time. There are some successful cases like amiR-F as mentioned. However, given the complexity of the regulatory network and interactions among miRNA and TFs, could the author discuss about what researchers should be aware of when engineering amiRNA to precisely control a trait/phenotype and minimize the side effects. Will some modeling based on co-expression network help more precise engineering?

minor comments:

(1) “Many plant miRNAs are evolutionarily conserved, and play vital role in the regulation of numerous essential developmental processes ”

please list out a few examples of the developmental processes that miRNA involved.

(2) “Current studies regarding the role of miRNAs in plant development are increasing significantly ”

What is the supporting evidence for this? Maybe get the number of publications on this topic by years on NCBI will be helpful.

(3) “MiRNAs are encoded in many loci in the intergenic regions of the plant genome” ‘’2.3. MiRNAs regulate members of transcription factors ”

MiRNA should be miRNA

(4) “In recent studies, it has been observed that sequence binding of miR156 and miR157 with their targets, SQUAMOSAPROMOTER BINDING PROTEIN-LIKE 3 (SPL3) gene at MRE region renders translational repression rather than degradation of the transcript ”

Be specific. Mention MRE is on 3’UTR.

(5) “In rice, 46 genes are predicted to be the targets of new miRNAs, while 16 genes encode the TF [32]. Similarly, it was found that 12 out of 26 miRNA families in maize regulate TF genes [18]. ”

Please mention the predictions are based on what types of analysis.

(6) “A recent study suggested that within the eight-member of miR156 family (miR156 a-h), miR156a and miR156c plays a leading role in determining the flowering time in Arabidopsis [64]. ”

Redundancy in this Sentence. Can be written as ‘ The eight-member miR156 family (miR156 a-h), miR156a and miR156c plays a leading role in determining the flowering time in Arabidopsis [64].’

Also “MiRNAs regulate members of transcription factors ” can be written as “miRNAs regulate transcription factors”

Please check for similar cases throughout the text.

(7) Use the same font for the figure legend of Figure2. There are also some small mistakes in the figure legend like “other factors such as sugar (HXK1) accumulation ”

(8) “4.1. SAM development ” Please spell out SAM in the title

(9) Section 4 is organized by the specific roles of miRNAs in Inflorescence development. I wonder if it possible to apply the same organization to section 3. The subtitles in section3 are just about miRNA families and is not quit helpful.

(10) “Chuck et al. showed that the wheat domestication gene Q encodes an AP2-like TF gene orthologous to IDS1. ” This is not an appropriate topic sentence for the paragraph. A topic sentence should be a summary of the paragraph. Please also check other paargraphs.

Reviewer 3 Report

It is unclear if the review manuscript is written on the basis of thorough and comprehensive study of the authors on plant miRNA biology. References in the manuscript are not up to date, and many of key study in plant miRNA research is missing in the review. The authors also incorrectly cite references and the required references are missing in many texts. Additionally, the authors often conclude and generalize their points without detailed description of the results published, which can mislead the readers. I also found many texts have grammatical errors and misspelling and in general, the manuscript seems to be written in poor English and so does not read well. Taken together, I strongly recommend the authors to thoroughly carry out scientific and editorial revision on their manuscript. Examples of my concerns are listed below.

1. Page 5. 'Overexpression of miR156 presents prolonged vegetative phase and late-flowering phenotypes [45]’

--> The cited study truly reported the heterochronic change in maize caused by miR156 overexpression, but I am not sure if the study presented miR156 overexpression causes delayed flowering in maize. Moreover, for the generalized conclusion, citing the maize study alone doesn’t seem sufficient. So the authors should tighten this conclusion or add additional references of studies in Arabidopsis and the other plants that were recently published.

2. Page 5. ‘miR156 and miR172 are the most ancient miRNAs’

--> References?

3. Page 5. ‘It is well known that miR156 regulated SPL genes display altered phenotypes such as the different number of leaves, semi-dwarfed size and retain longer vegetative phase [47].’

--> The authors should clarify that the study presents the case found in tomato. If the authors want to generalize this point, additional references are required. Moreover, the text should be corrected because miR156-regulated SPL genes does not display altered developmental phenotypes while the altered regulation of miR156-targeted SPLs can cause such phenotypes.

4. Page 5. ‘Similarly, miR390 and miR393 are involved in controlling the auxin signaling pathway to influence such actions of plants [52].’

--> The cited reference presents a role of miR167 in plant fertility. I found similar examples in the manuscript, and the authors should check their citation more carefully.

5. Page 5. ‘Latest studies demonstrated that some miRNAs act as key a regulator of auxin signaling pathway and directly or indirectly affect floral transition in different plant species [51].’

--> The authors cited a review paper. I recommend the authors search, study and cite the original research articles to guide readers directly to the studies. Similar issues can be found additionally in the text. And ‘as key a regulator’ should be ‘as a key regulator’.

6. Page 4. ‘The broad role of miRNAs in plant development is evident from the phenotypes of miRNA biogenesis defective mutants, miR168 regulates the expression of AGO genes, especially miR168b-regulated AGO1 plays crucial role in plant development [34], since miR168 is involved in regulating the key component of RISC, any variation in miR168 level has potential effects on the regulatory action of other miRNAs.’

--> Poor English writing. Must be revised.

7. Page 5. ‘Overexpression of miR393 demonstrated altered expression in Arabidopsis, Medicago truncatula and Oryza sativa by providing tolerance against stress responses such as low temperature, salinity and drought conditions’

--> Overexpression of miR393 demonstrated altered expression of what?

Round 2

Reviewer 1 Report

no more comments.

Author Response

Thank you for your comments.

Reviewer 3 Report

The authors have put efforts to address the concerns I raised in the previous review. However, I still find the same issues in the current version. I would like to make clear that the comments I made were the examples of problems that can be generally detected in the entire text. But the authors tried to correct only the part I pointed out. Hence I am not sure and satisfied if the authors thoroughly performed the scientific and editorial revisions throughout the manuscript. So I significantly concern if the manuscript satisfies a basic standard of review paper for the readership of GENES and strongly recommend to reject the manuscript. Followings are the examples of problems still detected.

<Major>

Examples of misleadings

1. Page 2. 'miRNA biogenesis is well understood in animal miRNAs. However, this process is still not very much familiar with respect to plants.' -> 'What is the scientific foundation of the authors' arguement? If the authors consider this true, they should present a summary (or figure) comparing the miRNA biogenesis between animals and plants and showing what is missing in plants.

2. Page 7. 'The first group consists of SPL3, SPL4 and SPL5, significantly influence the phase change and floral transition [79]' -> The authors cited the paper published in 2011 for the text. But the study more recently published shows that knocking out all three SPL3/4/5 using CRISPR does not affect the development of Arabidopsis (Xu et al 2016 PLoS Genet). So the authors' arquement can mislead the readers.

Examples of inappropriate and inadequate references

3. Page 7. ',selectively regulates the expression of miR156 targets [77]' -> In addition to the cited paper, there is one more paper published in the same issue (Yang et al. 2013 eLife). 

4. Page 7. 'A study revealed that glucose-sensing enzyme Hexokinase 1 (HXK1) is required for the upregulation of miR156A and miR156C expression in low sugar conditions [53]' -> Wrong citation. The reference does not present any of the results corresponding to this point.

5. Page 5. ', further complexity arises by AP2-like proteins....to fine-tune the flowering mechanism [92,95]' -> Wrong citation. Reference 92 is about AGO1 and miR166/165. Reference 95 is about the role of miR156 in the juvenility of the perennial plant Cardamine flexuosa.

6. Page 8. 'MiR390 affects flowering time through prolonging the juvenile phase, as a result of acquisition of the competence to flower [99]' -> The reference is a review article.

7. Page 8. 'In Arabidopsis, ta-siRNA-defective mutants showed accelerated juvenile to adult phase transition, due to an increased level of ARF3 and ARF4 [100]' -> The authors cited a review article.

Examples of editorial points

8.  Page 2, Line 58. SAM -> shoot apical meristem (SAM)

9. Page 6, Line 253. SPL -> SQUAMOSA PROMOTER BINDING PROTEIN LIKE (SPL)

10. Page 6, Line 260. MADs-box -> MADS-box

11. Page 8, Line 321. TOC1 and TOC2 -> TOE1 and TOE2

12. Page 8, Line 328. Ap2-like protein -> AP2-like protein

There are more issues throughout the manuscript while I haven't commented them here (because they are too many).  
